# High Throughput Sediment DNA Sequencing Reveals Azo Dye Degrading Bacteria Inhabit Nearshore Sediments

**DOI:** 10.3390/microorganisms8020233

**Published:** 2020-02-09

**Authors:** Mei Zhuang, Edmond Sanganyado, Liang Xu, Jianming Zhu, Ping Li, Wenhua Liu

**Affiliations:** 1Guangdong Provincial Key Laboratory of Marine Biotechnology, Institute of Marine Science, Shantou University, Shantou 515063, China; mei.zhuang@gtiit.edu.cn (M.Z.); xl9468zm@163.com (L.X.); liping@stu.edu.cn (P.L.); 2School of Marine Science and Technology, Harbin Institute of Technology at Weihai, Weihai 264209, China; 19b929041@stu.hit.edu.cn

**Keywords:** azo dye-degrading bacteria, nearshore sediment, high throughput sequencing, azoreductase genes, biodegradation

## Abstract

Estuaries and coastal environments are often regarded as a critical resource for the bioremediation of organic pollutants such as azo dyes due to their high abundance and diversity of extremophiles. Bioremediation through the activities of azoreductase, laccase, and other associated enzymes plays a critical role in the removal of azo dyes in built and natural environments. However, little is known about the biodegradation genes and azo dye degradation genes residing in sediments from coastal and estuarine environments. In this study, high-throughput sequencing (16S rRNA) of sediment DNA was used to explore the distribution of azo-dye degrading bacteria and their functional genes in estuaries and coastal environments. Unlike laccase genes, azoreductase (*azoR*), and naphthalene degrading genes were ubiquitous in the coastal and estuarine environments. The relative abundances of most functional genes were higher in the summer compared to winter at locations proximal to the mouths of the Hanjiang River and its distributaries. These results suggested inland river discharges influenced the occurrence and abundance of azo dye degrading genes in the nearshore environments. Furthermore, the *azoR* genes had a significant negative relationship with total organic carbon, Hg, and Cr (*p* < 0.05). This study provides critical insights into the biodegradation potential of indigenous microbial communities in nearshore environments and the influence of environmental factors on microbial structure, composition, and function which is essential for the development of technologies for bioremediation in azo dye contaminated sites.

## 1. Introduction

The presence of azo dyes in marine environments may pose significant ecological risk since they are highly recalcitrant and toxic [1]. Azo dyes are extensively used in textile, paper, and food industries where they are often discharged into the environment without prior treatment. It is estimated that the annual global production of azo dyes is 420,000 tons of which up to 15% is lost in the effluent [2]. Several studies detected azo dyes and their potentially toxic aromatic amines in industrial effluent and rivers at concentrations ranging from 0.05 to 316 µg L^−1^ [3]. Hence, current research has focused on the development of strategies for removing azo dyes from industrial effluent [4,5] and contaminated environments [6].

Azo dyes and their metabolites are highly toxic, carcinogenic, and mutagenic [1]. Exposing *Silurana tropicalis* (Western clawed frog) larvae to water and sediment containing Disperse Yellow 7 at environmentally relevant concentrations of up to 22 µg L^−1^ and 209 µg g^−1^, respectively resulted in a reduction in tadpole survival and an increase in malformations [7]. Sudan Red G was shown to decrease the survival of *Pimephales promelas* (fathead minnow) larvae with an LC50 of only 16.7 µg L^−1^ [8]. In rats, environmentally relevant concentrations of Disperse Red 7 induced DNA damage in the liver and upregulated expression of genes involved in inflammation [9]. Therefore, there is need for a comprehensive understanding of the fate of azo dyes in aquatic environments.

Indigenous microbial communities play a crucial role in the biodegradation of azo dyes in contaminated environments. Microbial enzymes such as azoreductase, laccase, and peroxidase are often involved in the initial reductive cleavage of the chromogenic azo bond (–*N*=*N*–) to produce lower molecular weight aromatic amines [10]. These potentially toxic intermediates are often characterized by thermally stable functional groups such as aniline, naphthalene, and benzidine [11]. Several studies have investigated the catabolic pathway and characterized the genes encoding the enzymes involved in the transformation of the aromatic nucleus of azo dyes (e.g., aniline, naphthalene, benzidine, and benzene) [12,13]. For example, previous studies investigated the distribution of microbial functional genes involved in PAH degradation in contaminated soils [14], aquifers [15], mangrove sediments [16], estuarine sediments [17], and coastal sediments [18]. In contrast, there is little information on the environmental distribution of microbial genes that encode proteins involved in the transformation of higher molecular weight azo dyes such as Acid Orange 7, Disperse Red 7, and Direct Blue 2B.

A comprehensive characterization of microbial functional diversity is essential for a better understanding of contaminated site bioremediation [19,20,21]. However, most studies on azo dye degradation focused on isolation, identification, and characterization of culturable bacteria [22,23]. It is estimated that only 0.1–1% microorganisms are culturable using conventional methods. Hence, culture methods do not provide adequate information on the biodegradation potential of the indigenous microbial communities in the contaminated environments [24]. Culture independent techniques such as high throughput environmental DNA (16S rRNA) sequencing have been extensively used to evaluate the distribution and diversity of functional genes in contaminated sites [25]; thus, providing a critical data on the potential of the environment to naturally eliminate the contaminants [13]. Several studies have demonstrated the utility of 16S rRNA in identification of novel enzymes or/and evaluation of the prevalence and distribution of contaminant-degrading microorganisms [26]. Thus, analysis of nearshore sediments can provide critical insights on the microbial functional diversity of azo dye degrading bacteria in estuaries and coastal environments.

The region surrounding Shantou Bay contributes more than 10% global production of lingerie and toys, industries which consume large volumes of azo dyes [6]. In the present study, we examined the distribution and diversity of azo dye degradation genes in nearshore sediment samples for which the physicochemical properties and metal pollution levels (i.e., salinity, total organic carbon (TOC), NO^3−^, NO^2−^, NH_4_^+^, PO_4_^3−^, SO_3_^2−^, Cu, Cr, As, Pb, Hg, and Zn) were characterized in our previous study [27]. These samples were screened using a combination of 16s rRNA sequencing and the Kyoto Encyclopedia of Genes and Genomes (KEGG) to predict the abundance of the functional genes. This enabled us to (i) evaluate the microbial functional structure in nearshore sediments; (ii) explore temporal variation in prevalence and distribution of azo dye degradation related functional genes; and (iii) assess the link between local environmental factors (e.g., salinity, total organic carbon (TOC), and metals) on diversity and abundance of the functional genes.

## 2. Materials and Methods

### 2.1. Sample Collection and Chemical Analysis

A total of 16 sediment samples were collected from eight sites on 5 November 2016 and 19 August 2017, respectively (Figure 1). The sampling sites were located on the mouth, distributary, and adjacent coastal areas of Rongjiang and Hanjiang Rivers in eastern Guangdong. Approximately 1 kg sediment was collected and placed in sterile polypropylene tubes at each sampling site from the top 10-cm seabed in triplicate. The sediment samples were then transported to the laboratory in an icebox.

Details of the procedures for assessing the physicochemical properties and metal concentration in these samples was provided in our previous study [27]. Briefly, inorganic nutrients (NO^3−^, NO^2−^, NH_4_^+^, PO_4_^3−^, and SO_3_^2−^) were determined using the “the specialties for oceanography survey” standard procedures (GB17378.4e2007, China) while salinity and TOC were analyzed using a smarTROLLTM multiparameter system (In-Situ Inc., Fort Collins, CO, USA) and an elemental analyzer (Multi N/C 2100, Germany), respectively. The metal (Cu, Cr, As, Pb, Hg, and Zn) concentrations were analyzed using inductively coupled plasma optical emission spectrometry (ICP- OES) and atomic fluorescence spectrometry techniques based on our previous method (Shi et al. [28]). Duplicate samples, method blanks, and a standard reference material obtained from the Chinese Academy of Measurement Sciences (GBW 07401) were used for quality control and assurance.

### 2.2. DNA Extraction and High-Throughput Sequencing

The microbial genomic DNA extraction from sediment samples and the polymerase chain reaction (PCR) were conducted according to our previous study [27]. Briefly, an OMEGA E.Z.N.A.^®^ Soil DNA Kit (Omega Bio-Tek, Norcross, GA, USA) was used to extract sediment DNA following the manufacturer’s protocols. The forward primer 515F and the reverse primer 806R were used to amplify the V4 region of the 16S rRNA gene. Following PCR, conversion of DNA fragments from jagged ends to blunt ends using T4 DNA polymerase, Klenow Fragment, and T4 Polynucleotide Kinase, addition of ‘A’ base to each 3′ end and then the removal of short fragments using Ampure beads, high throughput sequencing of the sediment DNA was conducted using Illumina HiSeq2500 platform (Beijing Genomics Institute, Wuhan, China). The raw fastq files were quality filtered and demultiplexed using USEARCH, the subsequent reads were cleaned by removing adapter pollution and low-quality reads, and the read pairs were merged using FLASH and then analyzed using UPARSE to obtain the operational taxonomic units (OTUs) at 97% similarity. Representative sequences were classified using the Ribosomal Database Project (RDP) classifier (v. 2.2) that was trained using the Greengenes database.

### 2.3. PICRUSt Approach

A literature search on SCOPUS and Web of Science was used to identify genes that encode enzymes involved in degradation of azo dyes, naphthalene, benzidine, benzene, and aniline. Phylogenetic Investigation of Communities by Reconstruction of Unobserved States (PICRUSt), a predictive exploratory tool, was then combined with the Kyoto Encyclopedia of Genes and Genomes (KEGG) ortholog classification to predict functional metagenomes from the 16S rRNA gene datasets of each sediment sample [29,30]. In this study, PICRUSt was used to explore the functional profiles of the bacterial communities according to the online protocol (http://picrust.github.io/picrust/).

### 2.4. Data Analysis

The experiments were performed in triplicate and the results were expressed as mean ± standard error. For the analysis of data, one-way analysis of variance (ANOVA) with Tukey–Kramer multiple comparisons test was used. The “*p* value” ≤ 0.05 indicated the readings are significant.

### 2.5. Data Availability

The sediment DNA sequences (16S rRNA genes and sequence reads) were deposited in the NCBI sequence read archive and the accession code is SRR9166248-9166255.

## 3. Results and Discussion

A total of 48,286 OTUs were found in 16 nearshore sediment samples, ranging from 342 (H2) to 7703 (H8). The coverage was over 97% for the all sediment samples. The diversity of sediment bacteria from this study was high with an average Shannon index of over 6.00 compared to the average Shannon index of 3.27 reported in the South China Seas, respectively [31]. The diversity indices of the present study were similar to those obtained in samples collected the following year in our previous study suggesting there was no apparent change in microbial diversity over the time period [27]. Proteobacteria was the dominant phylum in all samples with the average relative abundance of 46.17%. The top 20 phyla occupied approximately 97% of the OTUs (Figure 2a). At the class level, Gammaproteobacteria and Deltaproteobacteria, belonging to Proteobacteria, were two dominant groups with relative abundance of 20.26% and 16.04%, respectively (Figure 2b). Those results were consistent with the previous study in the same sampling area at different seasons, with two main class groups of Gammaproteobacteria and Deltaproteobacteria [27]. Moreover, several studies in nearshore sediment from different regions found similar results [32,33]. However, Anaerolineae at NH2 and Alphaproteobacteria at AH2, which were collected in same sample site from different seasons, were the most dominant classes, with relative abundance of 18.68% and 29.55% respectively, higher than Gammaproteobacteria and Deltaproteobacteria. While at NH7, Clostridia was the top group, followed by Bacteroidia, with relative abundance of 15.73% and 14.66%, respectively. Previous studies found that Anaerolineaceae (class Anaerolineae) abounded in organic-rich environments, such as anaerobic methanogenic sludge [34], or sediment [35], are possibly involved in organic matter mineralization process [36]. Bacteroidetes (class Bacteroidia) are normally associated with fermentative metabolism of labile high molecular weight organic matter, particularly under condition of sulfate-reducing or high nutrient concentrations [37]. Previous studies found Bacteroidetes were dominant in a hydrolysis acidification reactor [38], anaerobic sludge reactor [39], and a continuous reactor [40] during treatment of azo dyes suggesting these bacterial species were probably critical for maintenance of the bioreactors or possibly the decolorization of azo dyes. Interestingly, Wolińska et al. [41] found Bacteroidetes were a sensitive biological indicator of agricultural soil usage providing further evidence on their role in degradation of complex organic contaminants.

### 3.1. Spatial Distribution of Azo Dye-Related Degradation Genes

However, there is currently no data on the diversity, abundance, and distribution of azo dye-related degradation genes in the environment. However, previous studies have shown that functional genes are a powerful biomarker for determining the potential of indigenous microbial communities to degrade xenobiotic compounds [13]. In this study, in addition to azoreductase (*azo*R) gene, 12 genes associated with azo dye and naphthalene degradation were identified: laccase, *nah*Ab, *nah*Ac, *nah*Aa, *nah*C, *nah*D, *nah*P, *nah*F, tyrosinase, glutamine synthetase, cytochrome c peroxidase, and NADH peroxidase. Their relative abundances and diversities were estimated using KEGG and PICRUSt from each sampling site. Figure 3 shows that the functional genes associated with azo degradation were ubiquitous in the nearshore sediments, and the relative abundance and OTUs of the bacteria containing the genes varied with location. The relative abundance of all the genes were highest at H5 and H7 in winter, except *nah*P (H2) and cytochrome c peroxidase (H1). In summer, the relative abundances of all genes were highest at H2 and H5, except for laccase (H7) and *nah*P (H3). H2, H5, and H7 were located close to the mouths of the Hanjiang River and its distributaries. Thus, these results suggest that inland river discharges were probably influencing the occurrence and abundance of pollutant degrading genes in coastal and marine environments [42]. Similar results have been reported on the prevalence of PAH degradation functional genes in contaminated mangrove sediments [16], marine sediment [43], and tidal flat sediments [18]. The relative abundancies and diversities obtained through 16s rRNA sequencing, employed in this study, do not differentiate between genetic potential and actively expressed genes. Hence, the high abundance of azo dye degradation-related genes suggests microbial communities in coastal and estuarine sediments had potential of facilitating the degradation of azo dyes in these environments. To better understand the microorganism contributing to the degradation of azo dyes in coastal and estuarine sediments, comparative transcriptomics can be used to retrieve and sequence genes expressed during an active metabolic process [44]. Furthermore, systematic studies on the effect of azo dye gradients on azo dye-related degradation genes are required to assess the role of azo dyes on the proliferation and prevalence of azo dye-related degradation genes in marine environments.

The naphthalene degradation functional genes (*nah*Ab, *nah*Ac, *nah*Aa, *nah*C, *nah*D, *nah*P, and *nah*F) are mainly involved in the downstream degradation of naphthalene-based dyes such as Acid Orange 7, Reactive Black 5, and Congo Red. The relative abundance of naphthalene degradation functional genes were high at most sampling sites probably because they are not exclusive to the degradation of azo dyes but involved in degradation of polycyclic aromatic hydrocarbons [12,14,16]. The naphthalene base is commonly found in other organic pollutants in marine environments such as polycyclic aromatic hydrocarbons [18]. Similarly, glutamine synthetase gene, which has been shown to play a critical role in degradation of aniline, had the highest diversity in all samples [45]. Cytochrome c peroxidase genes were evenly distributed across all sampling sites ranging 0.5% to 5.8%, except at H7. However, the *azo*R genes are involved in the upstream degradation and are probably more specific to azo dye degradation; thus, they are probably a better functional marker of azo dye degrading bacteria.

### 3.2. Seasonal Variations in Azo Dye-Related Degradation Genes

Several studies have shown seasonal variations in the discharges from inland rivers, stormwater runoff, and wastewater treatment plants cause spatiotemporal trends in physicochemical properties of coastal and estuarine sediments such as organic nutrients, pH, salinity, and metals [46,47]. Together with oceanographic factors such as ocean currents changes in these physicochemical properties may cause shifts in microbial community structure [27,48,49] as well as co-regulate the prevalence and distribution of functional genes in coastal and estuarine sediments [50]. A previous study in the Lake Bosten, a brackish inland lake in an arid region in northwest China found that there was higher diversity in sediment microbial communities in winter than in summer [51]. In this study, a significant seasonal variation was observed in the relative abundance of *azo*R (*p* = 0.01), *nah*P (*p* = 0.01), glutamine synthetase (*p* = 0.007), and NADH peroxidase (*p* = 0.01). The relative abundance of *azo*R genes increased in summer at all sites except at H7 where it decreased from 5.3% to 4.8%. A similar trend was observed for glutamine synthetase, with the relative abundance decreasing from 4.8% to 4.4% at H7. However, no such decrease at H7 was observed for *nah*P genes. Between H7 and H8 there was a barrier built that prevented the northward flow of discharges from the Rongjian River, which has previously been shown to be highly contaminated by metals [27]. However, the number of unique OTUs with the *azo*R gene decreased in summer by at least 50% at H1, H2, H3, H5, and H6 but increased at H7 and H8 by 84% and 66%, respectively. Coastal and marine infrastructural developments can change local environmental factors such as water-flow, light, and organic pollutants (e.g., antifouling agents from paints and polycyclic aromatic hydrocarbons from fuel combustion) resulting in localized shifts in microbial community structure, composition, and function [52]. A previous study showed that barriers shifted microbial communities by altering water flow and increasing woody detritus [53]. Our results suggest there is a need to further explore the effect of artificial structures on microbial functions to better understand the foundational processes driving urban marine ecology.

### 3.3. Distribution of Azo Reductase Genes in Bacteria

The distribution of *azo*R genes varied between bacteria communities and was detected in diverse groups of bacteria (Figure 4). Seasonal variations in the relative abundance of *azo*R-containing orders was observed. Pirellulales, Rhizobiales, and Marinicellales were the top three orders for sediment sample (NH1-NH8) collected in November, with relative abundance of 4.08%, 3.11%, and 3.02%, respectively. For the samples collected in August (AH1-AH8), Pirellulales, Rhizobiales, Burkholderiales, Marinicellales, and Pseudomonadales were the top five orders with the relative abundance of 5.97%, 3.39%, 3.29%, 2.78%, and 2.63%, respectively. Previous studies have shown that organic pollutant degrading genes can be horizontally transferred between different bacteria species [43]. The relative abundance of *azo*R-containing Proteobacteria and Planctomycetes were higher in summer than in winter as observed in previous studies [54]. These changes in microbial community composition were probably caused by the seasonal changes in local environmental factors (day length, pH, salinity, and temperature) [55]. Alternatively, the seasonal variation may indicate there was horizontal gene transfer between bacteria of different orders. Horizontal gene transfer has been shown to improve the adaptation of the bacteria to environmental stressors such as metals [56], antibiotics [57], and polycyclic aromatic hydrocarbons [43].

### 3.4. Effect of Local Environmental Factors on Distribution of Azo Dye-Related Genes

Understanding the effect of local environmental factors is of great importance in microbial ecology [58]. Our previous study showed that the sediments collected for this study were impacted by inorganic nutrients and metals (Table 1) [27]. The metal concentration was higher at H5, H7, and H8 which were proximal to the mouth of Rongjiang and Hanjiang Rivers (and its distributary). These results suggest river discharge may have contributed to metal pollution. A similar trend was observed for NO^3−^ and NO^2−^.

Table 2 shows the Pearson correlations of the environmental factors (salinity, TOC, and heavy metals) and functional gene abundance. There was a significant negative correlation between the relative abundance of *azo*R genes and TOC (*p* < 0.05), Cr (*p* < 0.05), and Hg (*p* < 0.05). *Nah*Ab also showed significant negative correlation with TOC and Cr. Similarity, *nah*P had significant negative correlation with Cu, Zn, As, and Hg. Previous studies showed that metals inhibited bacteria [59]. However, *nah*C had significant positive correlation with Cu and Pb. Except Pb, all heavy metals showed significant negative correlation with the relative abundance of glutamine synthetase. There was a significant negative correlation between salinity and laccase.

The Pearson correlation among azo dye degrading related genes (Table 3) showed *azo*R genes had a significant positive correlation with *nah*Ab, *nah*Ac, *nah*Aa, *nah*D, tyrosinase and glutamine synthetase. These results suggest *azo*R and the naphthalene degradation genes probably evolved closely [60]. Except *nah*P and *nah*F, there were significant positive correlation between *nah*Aa, *nah*Ab, *nah*Ac, *nah*C, and *nah*D. Meanwhile, tyrosinase and glutamine synthetase showed significant positive correlation with *nah*Aa, *nah*Ab, *nah*Ac, *nah*D, *nah*P, and *nah*F. Previous studies showed that naphthalene degradation genes often co-exist in the same bacterial hosts [61]. However, cytochrome c peroxidase had significant negative correlation with *nah*P and *nah*F.

### 3.5. Environmental Implications

A better understanding of the prevalence and distribution of biodegradation functional genes as well as the effect of local environmental factors can be important for setting long-term environmental management and conservation goals. Functional genes have been shown to indicate potential of an environmental system to biodegrade an organic pollutant [12,13]. Hence, the presence of diverse microbial communities with biodegradation functional genes following a contamination event indicate the adaptability of the environment to contamination in addition to potential horizontal gene transfer across species [13,17]. Microbial diversity is particularly important considering synthetic azo dyes are structurally diverse and bacteria vary in their tolerance to these dyes [62]. The frequent detection of *azo*R in the nearshore sediments suggest possible contamination by azo compounds such as synthetic azo dyes and natural organic pigments. Furthermore, the prevalence of naphthalene degradation genes in the nearshore sediments suggest there could be significant PAH contamination in the region.

Since the current study used high throughput sediment DNA sequencing based on 16S rRNA, definitive conclusion on the adaptability of the nearshore systems to azo dye contamination could not be drawn [30]. The PICRUST approach used in this study only predicted the microbial community metagenomic content [63], based on our previous genome sequences [27]. Although several studies have shown that *azo*R and laccase genes are the key genes involved in the upstream degradation of azo dyes, microorganisms sometimes have unique catabolic pathways that involve novel genes that are not listed on genomic databases [30,63]. Therefore, the current study adequately generated the hypotheses (i) *azo*R can be used as a biomarker for the biodegradation potential of synthetic azo dye in marine environments and (ii) prevalence and distribution of *azo*R genes indicates the adaptability of the nearshore environments to azo dye contamination. Hence, future studies should investigate the impact of synthetic azo dye gradients to distribution of *azo*R genes in marine environments using direct metagenomic and metatranscriptomic sequencing to better understand microbial functions and abundances.

## 4. Conclusions

The potential of in situ bioremediation can be determined by establishing the distribution microorganisms and biodegradation functional genes in the contaminated site. Marine environments are often treated as a crucial source of extremophiles that can be isolated, cultured, and developed for biotreatment of textile effluent and azo dye impacted soils. However, marine environments are also impacted by azo dye contamination. In this study, we found a high abundance of downstream degradation functional genes at all sites, indicating existence of native microorganisms essential for naphthalene degradation. However, the abundance of upstream degradation functional genes (*azo*R and laccase) varied between sampling sites. In addition to spatial factors, distribution of functional genes was shown to be influenced by local environmental factors. Significant negative correlation was observed between *azo*R and Cr and Hg, and *nah*P and Cu, Zn, As, and Hg. This suggest that contamination of sediments by metals may influence the biodegradation of synthetic azo dyes. In this study, TOC and salinity also significantly affected the distribution of some functional genes. Therefore, local environmental conditions may present a great challenge in development of in situ bioremediation strategies for marine environments contaminated by synthetic azo dyes.

The presence of functional genes involved in degradation of synthetic azo dyes in nearshore sediments indicates that indigenous microorganisms could be enriched for remediating contaminated sites. Since the distribution and functional diversity varied between sites, a successful in situ bioremediation approach might require careful optimization of environmental conditions as well as maintenance of microbial populations. Where critical microbes involved in biodegradation are absent, non-indigenous bacteria can be introduced. However, proliferation success of the non-indigenous bacteria depends on their ability to colonize the new environment. Alternatively, novel genetic engineering techniques such as CRISPR/Cas9 can be used as a tool for modifying indigenous bacteria. Understanding the potential of microbial communities to degrade organic pollutants, the spatial factors and local environmental factors influencing microbial community structure, composition, and function is critical for understanding adaptability of natural environments to contamination as well as designing effective in situ bioremediation strategies.

## Figures and Tables

**Figure 1 microorganisms-08-00233-f001:**
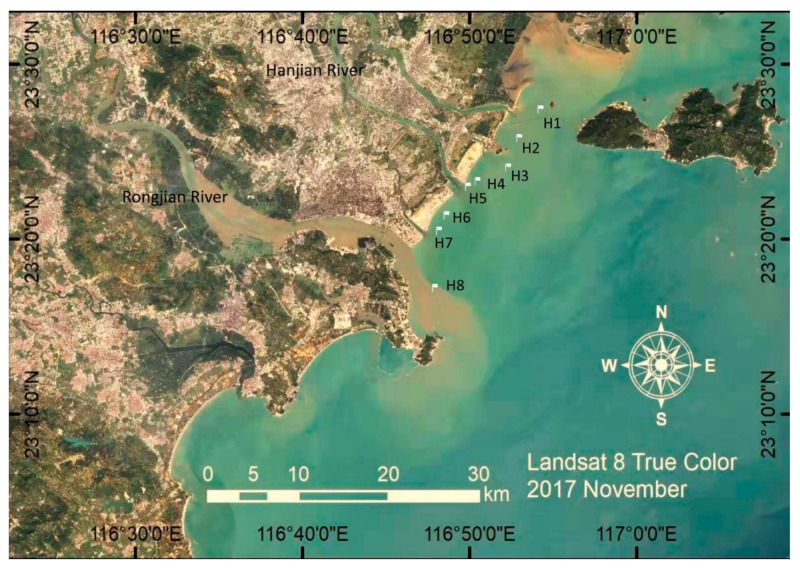
Sampling sites for nearshore sediments along the coastline of Shantou, China.

**Figure 2 microorganisms-08-00233-f002:**
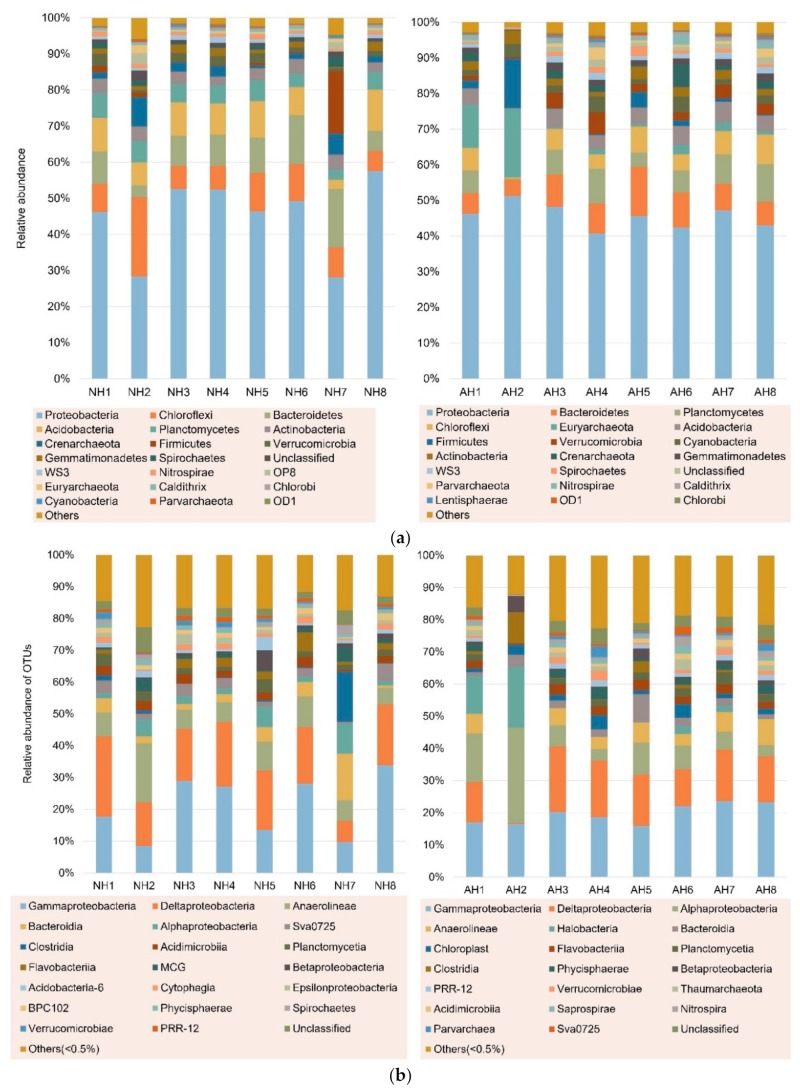
Taxonomic composition distribution histograms in each sample at (**a**) phylum and (**b**) class level.

**Figure 3 microorganisms-08-00233-f003:**
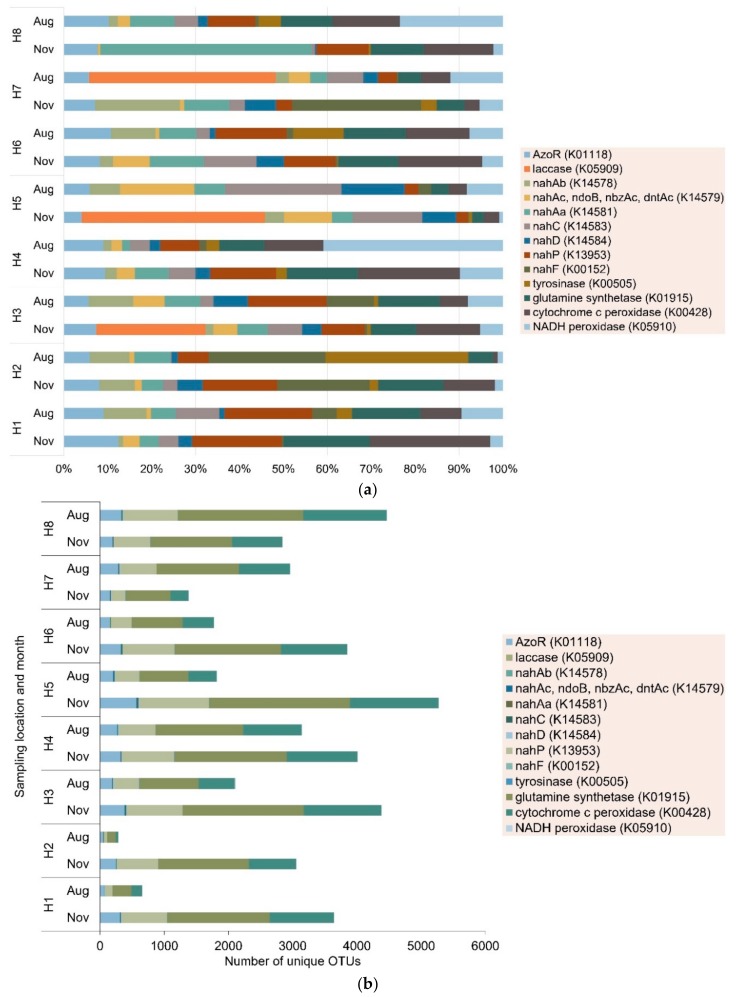
The (**a**) relative abundance and (**b**) operational taxonomic unit (OTUs) of the annotated functional genes involved in synthetic azo dye degradation.

**Figure 4 microorganisms-08-00233-f004:**
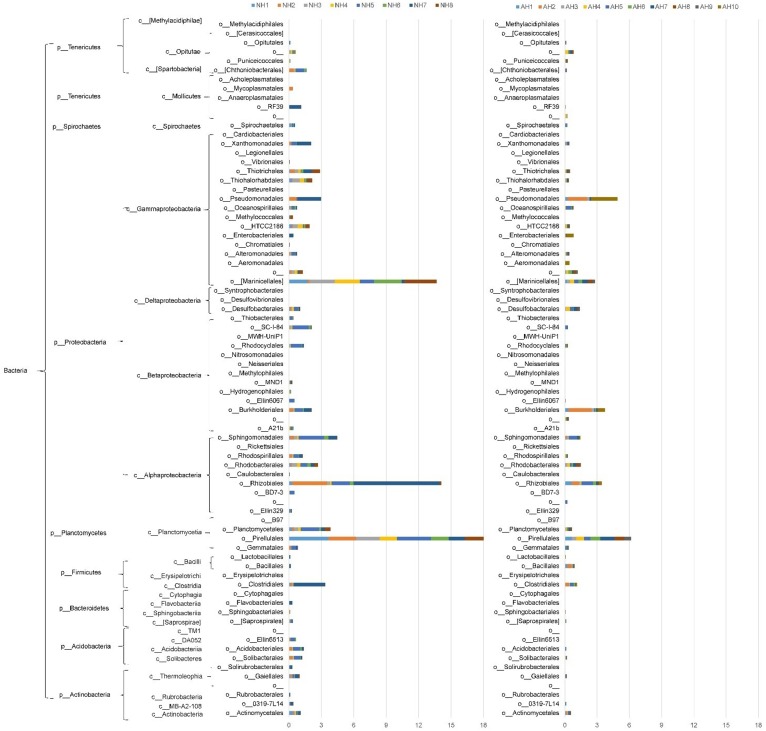
The distribution of *azo*R genes in diverse groups of bacteria.

**Table 1 microorganisms-08-00233-t001:** Physicochemical properties of sediment samples (adapted from Zhuang et al. [27]).

Sites	Salinity (‰)	TOC (g/kg)	Inorganic Nutrients (mg/L)					Metal Concentration in Sediment (mg/kg)					
			NO_3_^−^	NO_2_^−^	NH_4_^+^	PO_4_^3^	SO_3_^2^	Cu	Zn	Cr	Pb	As	Hg
H1	30.3	4.2	0.096	0.001	0.063	0.027	0.695	13.8	59.6	16.7	29.6	5.8	0.045
H2	20.4	6.5	0.335	0.009	0.085	0.028	1.382	20.4	76.1	24.3	53.0	8.3	0.090
H3	27.7	5.8	0.042	0.001	0.037	0.029	0.132	21.9	87.6	29.7	38.1	8.2	0.075
H4	30.4	9.0	0.122	0.004	0.08	0.018	0.538	35.0	109.0	36.8	53.7	10.1	0.080
H5	22.9	7.5	0.296	0.006	0.086	0.039	1.007	75.8	155.0	24.3	95.7	11.5	0.116
H6	20.5	2.0	0.421	0.009	0.104	0.043	1.866	10.3	51.2	16.7	24.0	5.9	0.046
H7	27.8	9.0	0.072	0.007	0.061	0.028	0.616	43.0	123.0	39.1	57.9	10.9	0.084
H8	29.6	9.3	0.316	0.042	0.142	0.057	0.866	37.6	135.0	52.5	43.1	9.0	0.097

**Table 2 microorganisms-08-00233-t002:** Pearson’s correlation coefficient (r) of azo dye degrading related genes and environmental parameters in sediment samples. (* *p* < 0.05; ** *p* < 0.01).

	*Azo*R	Laccase	*nah*Ab	*nah*Ac	*nah*Aa	*nah*C	*nah*D	*nah*P	*nah*F	Tyrosinase	Glutamine	Cytochrome	NADH
TOC	−**0.482 ***	0.036	−**0.448 ***	−0.292	−0.225	0.011	−0.359	−0.379	−0.240	−0.292	−**0.543 ***	0.031	−0.062
Cu	−0.286	0.112	−0.303	0.014	−0.072	**0.501 ***	−0.094	−**0.444 ***	−0.291	−0.325	−**0.504 ***	0.164	0.105
Zn	−0.375	−0.030	−0.390	−0.117	−0.093	0.266	−0.209	−**0.436 ***	−0.333	−0.329	−**0.524 ***	0.203	0.049
Cr	−**0.468 ***	−0.231	−**0.437 ***	−0.376	−0.142	−0.263	−0.394	−0.393	−0.249	−0.287	−**0.478 ***	0.170	−0.073
Pb	−0.262	0.143	−0.222	0.024	−0.080	**0.516 ***	−0.078	−0.334	−0.092	−0.198	−0.405	−0.072	−0.004
As	−0.379	0.118	−0.287	−0.119	−0.130	0.190	−0.190	−**0.448 ***	−0.208	−0.242	−**0.549 ****	−0.065	−0.133
Hg	−**0.480 ***	0.047	−0.308	−0.258	−0.113	0.033	−0.287	−**0.529 ***	−0.059	−0.299	−**0.650 ****	−0.238	−0.391
Salinity	0.199	−**0.553 ****	0.103	0.118	0.023	−0.278	0.153	0.134	−0.348	0.125	0.191	0.418	0.259

**Table 3 microorganisms-08-00233-t003:** Pearson’s correlation coefficient (*r*) between azo dye degrading related genes in sediment samples. (* *p* < 0.05; ** *p* < 0.01).

	*Azo*R	Laccase	*nah*Ab	*nah*Ac	*nah*Aa	*nah*C	*nah*D	*nah*P	*nah*F	Tyrosinase	Glutamine	Cytochrome	NADH
*Azo*R	1												
laccase	0.016	1											
*nah*Ab	**0.941 ****	−0.047	1										
*nah*Ac	**0.899 ****	0.137	**0.835 ****	1									
*nah*Aa	**0.854 ****	−0.026	**0.892 ****	**0.810 ****	1								
*nah*C	0.313	0.380	0.229	**0.590 ****	0.243	1							
*nah*D	**0.942 ****	0.044	**0.912 ****	**0.982 ****	**0.859 ****	**0.468 ***	1						
*nah*P	0.335	−0.197	0.358	0.159	0.251	−0.164	0.212	1					
*nah*F	0.177	−0.189	0.331	−0.100	0.206	−0.145	−0.004	0.403	1				
tyrosinase	**0.899 ****	−0.128	**0.820 ****	**0.622 ****	**0.783 ****	−0.020	**0.683 ****	**0.486 ***	**0.463 ***	1			
glutamine	**0.589 ****	−0.212	**0.616 ****	0.362	**0.446 ***	−0.023	**0.440 ***	**0.898 ****	**0.521 ***	**0.615 ****	1		
cytochrome	−0.072	0.152	−0.318	0.072	−0.287	0.220	−0.018	−**0.536 ***	−**0.768 ****	−0.422	−**0.456 ***	1	
NADH	0.016	0.057	−0.144	−0.069	−0.223	0.103	−0.085	−0.027	−0.172	−0.192	0.105	0.432	1

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
