# Peer review of "High Throughput Sediment DNA Sequencing Reveals Azo Dye Degrading Bacteria Inhabit Nearshore Sediments"

_microorganisms, 2020, doi:10.3390/microorganisms8020233_

Round 1

Reviewer 1 Report

The results presented in the manuscript titled "High throughput sediment DNA sequencing reveals azo dye degrading bacteria inhabit nearshore sediments" by Mei Zhuang et al.  are of considerable interest for understanding the processes occurring in microbial communities.
 In my opinion, the article can be published in its current form.  There are only small notes:
1. Lines 51-52.  The phrase "Indigenous microbial... contaminated environments", the word "critical" may be replaced by "crucial".

2. Line 152. Figure 2. Taxonomic composition distribution histograms in each sample at (a) phylum and (b) class
 level. Maybe I didn’t catch the subtleties of the classification, but it seems to me that the results presented in Figure 2 "Taxonomic composition distribution histograms in each sample at (a) phylum and (b) class level"  in the presented form do not carry a sufficient amount of various information.  And, probably, Figure 2 can be halved to the Class level.

3. Line 167. "H2, H5 and H7 were locate close to …" should be "H2, H5 and H7 were located close to..."

4. Lines 244-245. "Meanwhile, tyrosinase and glutamine synthetase showed significant positive correlation with between nahAa, nahAb, nahAc, nahD, nahP and nahF." Correlation with or between?

5. Conclusion. Lines 288-291. The sentence "The current  study considered the marine systems as impacted by synthetic azo dyes rather than as not only as a 
 source of microbes that are isolated and for use in other environments such as contaminated soils or textile effluent treatment processes but also as an environment impacted by azo dye contamination". It is too long and contains the repeated word "as".  This sentence should be rephrased, including part " that are isolated and for use in other".

6. Line 323. The APC was funded by XXX.  ?

7. Latin names should be in italic (e.g., lines 352, 487).

8. In the list of references in the names of articles, words should begin with a small letter, unless specifically required (e.g., citations 1, 19, 26, 35, 38, 40, 45, 47)

Author Response

The results presented in the manuscript titled "High throughput sediment DNA sequencing reveals azo dye degrading bacteria inhabit nearshore sediments" by Mei Zhuang et al.  are of considerable interest for understanding the processes occurring in microbial communities.

In my opinion, the article can be published in its current form.  There are only small notes:

Lines 51-52. The phrase "Indigenous microbial... contaminated environments", the word "critical" may be replaced by "crucial".

We replaced critical with crucial.

Line 152. Figure 2. Taxonomic composition distribution histograms in each sample at (a) phylum and (b) class level. Maybe I didn’t catch the subtleties of the classification, but it seems to me that the results presented in Figure 2 "Taxonomic composition distribution histograms in each sample at (a) phylum and (b) class level" in the presented form do not carry a sufficient amount of various information. And, probably, Figure 2 can be halved to the Class level.

Thank you for the comment. We replaced Figure 2b with the correct figure.

Line 167. "H2, H5 and H7 were locate close to …" should be "H2, H5 and H7 were located close to..."

Thank you for the comment. We have revised it to located.

Lines 244-245. "Meanwhile, tyrosinase and glutamine synthetase showed significant positive correlation with between nahAa, nahAb, nahAc, nahD, nahP and nahF." Correlation with or between?

Thank you for the correction. We removed between.

Conclusion. Lines 288-291. The sentence "The current study considered the marine systems as impacted by synthetic azo dyes rather than as not only as a source of microbes that are isolated and for use in other environments such as contaminated soils or textile effluent treatment processes but also as an environment impacted by azo dye contamination". It is too long and contains the repeated word "as". This sentence should be rephrased, including part " that are isolated and for use in other".

Thank you for the comment. The following correction was made: The potential of in situ bioremediation can be determined by establishing the distribution of microorganisms and biodegradation functional genes in the contaminated site. Marine environments are often treated as a crucial source of extremophiles that can be isolated, cultured, and developed for biotreatment of textile effluent and azo dye impacted soils.

Line 323. The APC was funded by XXX.?

Thank you for the comment. We edited the line to read The APC was funded by E.S..

Latin names should be in italic (e.g., lines 352, 487).

Thank you for the comment we have revised the references.

In the list of references in the names of articles, words should begin with a small letter, unless specifically required (e.g., citations 1, 19, 26, 35, 38, 40, 45, 47)

Thank you for the comment we have revised the references.

Reviewer 2 Report

Evaluated paper is interesting and is well and logically prepared.  The study provides critical insights into the biodegradation potential of indigenous microbial communities in nearshore environments and the influence of environmental factors on microbial structure, composition, and function which is essential for the development of technologies for bioremediation in azo dye contaminated sites. However some major correction should be performed by Authors before paper acceptation.

Line 23: should be: the highest

Line 65: I suggest to add the following citation here:  WoliÅ„ska A., GaÅ‚Ä…zka A., Kuźniar A., Goraj W., JastrzÄ™bska N., GrzÄ…dziel J., StÄ™pniewska Z. 2018. Catabolic fingerprinting and diversity of bacteria in Mollic Gleysol contaminated with petroleum substances. Applied Sciences 8: 1970.

Line 80: should be: Zhuang et al. [32]

Line 90: should be: placed in sterile…

Line 96: should be: [32]

Line 99: should be: Following PCR….

Line 118: should be: data and have been…

Line 134: should be: Proteobacteria

Line 147: please add the following information: [41], whereas WoliÅ„ska et al. […] found this phyla as a sensitive biological indicator of agricultural soil usage revealed by culture independent approach.

Wolińska A., Kuźniar A., Zielenkiewicz U., Izak D., Szafranek-Nakonieczna A., Banach A., Błaszczyk M. 2017. Bacteroidetes as a sensitive biological indicator of agricultural soil usage revealed by culture independent approach. Applied Soil Ecology, 119: 128-137.

The important mistake that must be corrected is the fact that in the Results and Discussion section there is lack of sub-point regarding chemical characteristic of studied material. In the 3.4. section Authors described correlations among azo-dye related genes with environmental factors (TOC, salinity, selected metals) but the reader nothing know about the content of this factors in the studied material (I suggest to add Table). In this context also Material and methods section should be supplemented by information about measuring way of studied environmental factors.

Also an information about pH should be added, as pH is one of the most important environmental factor influencing on microbial structure.

Also References style is not accepted in the current form and should be rewritten according  journal demands ( please read the Guide for authors).

Author Response

Evaluated paper is interesting and is well and logically prepared.  The study provides critical insights into the biodegradation potential of indigenous microbial communities in nearshore environments and the influence of environmental factors on microbial structure, composition, and function which is essential for the development of technologies for bioremediation in azo dye contaminated sites. However, some major correction should be performed by Authors before paper acceptation.

Thank you for the insightful comments. We have made the suggested corrections to improve the quality of the manuscript.

Point 1: Line 23: should be: the highest

Thank you for the comment. We revised the sentence as follows: The relative abundances of most functional genes were higher in the summer compared to winter at locations proximal to the mouths of the Hanjiang River and its distributaries.

Point 2: Line 65: I suggest to add the following citation here:  WoliÅ„ska A., GaÅ‚Ä…zka A., Kuźniar A., Goraj W., JastrzÄ™bska N., GrzÄ…dziel J., StÄ™pniewska Z. 2018. Catabolic fingerprinting and diversity of bacteria in Mollic Gleysol contaminated with petroleum substances. Applied Sciences 8: 1970.

Thank you for the comment. We have added the reference.

Point 3: Line 80: should be: Zhuang et al. [32]

Thank you for the comment. We have revised the reference.

Point 4: Line 90: should be: placed in sterile…

Thank you for the comment. We have added ‘in’.

Point 5: Line 96: should be: [32]

Thank you for the comment. We have revised the reference.

Point 6: Line 99: should be: Following PCR….

Thank you for the comment. We have removed the term polymerase chain reaction.

Point 7: Line 118: should be: data and have been…

We revised the sentence as follows: The experiments were performed in triplicate and the results were expressed as mean ± standard error. 

Point 8: Line 134: should be: Proteobacteria

We made the correction.

Point 9: Line 147: please add the following information: [41], whereas WoliÅ„ska et al. […] found this phyla as a sensitive biological indicator of agricultural soil usage revealed by culture independent approach.

Wolińska A., Kuźniar A., Zielenkiewicz U., Izak D., Szafranek-Nakonieczna A., Banach A., Błaszczyk M. 2017. Bacteroidetes as a sensitive biological indicator of agricultural soil usage revealed by culture independent approach. Applied Soil Ecology, 119: 128-137.

Thank you for the comment. The following correction was made: Previous studies found Bacteroidetes were dominant in a hydrolysis acidification reactor [45], anaerobic sludge reactor, and a continuous reactor [46] during treatment of azo dyes suggesting these bacterial species were probably critical for maintenance of the bioreactors or possibly the decolorization of azo dyes. Interestingly, Wolińska et al. [48] found Bacteroidetes were a sensitive biological indicator of agricultural soil usage providing further evidence on their role in the degradation of complex organic contaminants.

Point 10: The important mistake that must be corrected is the fact that in the Results and Discussion section there is lack of sub-point regarding chemical characteristic of studied material. In the 3.4. section Authors described correlations among azo-dye related genes with environmental factors (TOC, salinity, selected metals) but the reader nothing know about the content of this factors in the studied material (I suggest to add Table). In this context also Material and methods section should be supplemented by information about measuring way of studied environmental factors.

Thank you for the comments. We have revised the manuscript to add the details on the procedure for analyzing metals, inorganic nutrients, salinity, and TOC. We have added Table 1 as suggested by the reviewer.

Point 11: Also, an information about pH should be added, as pH is one of the most important environmental factor influencing on microbial structure.

We agree that pH might be an important factor influencing microbial structure. We did not measure the pH of the coastal sediments we sampled.

Point 12: Also References style is not accepted in the current form and should be rewritten according  journal demands ( please read the Guide for authors).

Thank you for the comment. We revised the reference style so that they meet MDPI style guide.

Round 2

Reviewer 2 Report

Thank you for the correction o the paper. The current version is much better and is suitable for publication.